# Role of miRNAs in Neurodegeneration: From Disease Cause to Tools of Biomarker Discovery and Therapeutics

**DOI:** 10.3390/genes13030425

**Published:** 2022-02-25

**Authors:** Bidisha Roy, Erica Lee, Teresa Li, Maria Rampersaud

**Affiliations:** 1Life Science Centre, Department of Biological Sciences, Rutgers University-Newark, Newark, NJ 07012, USA; 2Department of Pathology, Icahn School of Medicine, New York, NY 10029, USA; ericalee835@gmail.com (E.L.); tli110888@gmail.com (T.L.); m.rampersaud52495@gmail.com (M.R.)

**Keywords:** miRNAs, neurodegenerative disease, biogenesis, disease mechanisms, biomarkers and therapeutics

## Abstract

Neurodegenerative diseases originate from neuronal loss in the central nervous system (CNS). These debilitating diseases progress with age and have become common due to an increase in longevity. The National Institute of Environmental Health Science’s 2021 annual report suggests around 6.2 million Americans are living with Alzheimer’s disease, and there is a possibility that there will be 1.2 million Parkinson’s disease patients in the USA by 2030. There is no clear-cut universal mechanism for identifying neurodegenerative diseases, and therefore, they pose a challenge for neurobiology scientists. Genetic and environmental factors modulate these diseases leading to familial or sporadic forms. Prior studies have shown that miRNA levels are altered during the course of the disease, thereby suggesting that these noncoding RNAs may be the contributing factor in neurodegeneration. In this review, we highlight the role of miRNAs in the pathogenesis of neurodegenerative diseases. Through this review, we aim to achieve four main objectives: First, we highlight how dysregulation of miRNA biogenesis led to these diseases. Second, we highlight the computational or bioinformatics tools required to identify the putative molecular targets of miRNAs, leading to biological molecular pathways or mechanisms involved in these diseases. Third, we focus on the dysregulation of miRNAs and their target genes leading to several neurodegenerative diseases. In the final section, we highlight the use of miRNAs as potential diagnostic biomarkers in the early asymptomatic preclinical diagnosis of these age-dependent debilitating diseases. Additionally, we discuss the challenges and advances in the development of miRNA therapeutics for brain targeting. We list some of the innovative strategies employed to deliver miRNA into target cells and the relevance of these viral and non-viral carrier systems in RNA therapy for neurodegenerative diseases. In summary, this review highlights the relevance of studying brain-enriched miRNAs, the mechanisms underlying their regulation of target gene expression, their dysregulation leading to progressive neurodegeneration, and their potential for biomarker marker and therapeutic intervention. This review thereby highlights ways for the effective diagnosis and prevention of these neurodegenerative disorders in the near future.

## 1. Introduction

Neurodegenerative diseases such as Alzheimer’s disease (AD), Parkinson’s disease (PD), Huntington’s disease (HD), and Amyotrophic lateral sclerosis (ALS) are a group of age-dependent progressive disorders initiated by the loss of neurons that eventually lead to cognitive and movement disorders. These diseases are thought to be caused by alterations to protein-coding genes. These alterations arise from somatic genetic events that occur over long periods of time [1]. Noncoding RNAs comprise 95% of human cellular RNAs [2]. These noncoding RNAs participate in translational regulation and have thereby led researchers to seek an in-depth understanding of mRNA translation [3]. miRNAs are small, noncoding RNA molecules transcribed from RNA polymerase II and III. Their function is closely associated with gene regulation. The regulation of genes is controlled during the stage of post-transcription where its goal is to bind its target mRNA and negatively regulate its expression to inhibit the production of proteins. A strand of mature miRNA is formed and included in the effector complex where it acts as a post-transcriptional regulator with its target mRNA. The way the inhibition of producing proteins is achieved is dependent on how well the base pairs bind together (complementarity), which in turn activates one of two mechanisms-degradation of the mRNA or the blockage of translation. These small but impactful molecules make up at least 1% of the human genome, and the dysfunctional manner of miRNAs has been closely associated with many known diseases.

miRNAs are 22 nucleotides long and fixed in the 3′ untranslated section. Each miRNA has a conserved region, known as the seed region. The seed region comprises an area between 2 to 8 nucleotides, numbered from the 5′ to 3′ ends of the miRNA sequence, and has a perfect or nearly perfect Watson–Crick complementarity with the 3′ UTR of the mRNA. In 1993 Lee and Feinbaum et al. discovered lin-4, a gene that modulated C.elegans development and coded for a small nonprotein-coding RNA [4]. After extensive research, it was found that small noncoding miRNAs played an important role in fine-tuning the genome [5]. Transcriptome usage in different combinations leads to the formation of complex cellular networks in the brain. This is achieved by modulating the expression of thousands of genes, contributing to various physiological processes [6]. Genome sequence variants associated with HD affected miRNA binding to the BDNF (Brain-derived neurotrophic factor) [7,8]. Changes in miRNA expression profiles in AD [9,10,11] and PD [12] patients are examples of miRNAs potentially involved in neurodegenerative diseases [13,14]. Therefore, understanding fundamental aspects of miRNA neurobiology and the possible clinical implications related to miRNA dysfunction are clearly important for many fields of neuroscience and neurological disorders. To date, the involvement of miRNAs in various physiologically relevant processes in neurodegeneration has been established [3,15,16]. Previously, several researchers confirmed diverse sources of miRNAs are present in the brain, some of which follow a very specific pattern of expression during its development [6,17,18]. These miRNAs serve as regulators of important biological pathways. Additionally, with the passing of years, a functional role for miRNAs in specific neurological processes is emerging, and their dysfunction has direct relevance for our understanding of neurodegenerative disorders [19].

In this review, we highlight the main progress in the field of miRNA research and its impact on neurodegeneration. These signs of progress include first, the research findings with respect to what is known regarding the disruption of proper miRNA biogenesis leading to neurodegeneration. Second is the identification of miRNAs that target specific disease genes. Third, we summarize the potential of some of these brain-enriched miRNAs in preclinical biomarker studies. Last, we focus on the promising use of these miRNAs as potential diagnostic and therapeutic agents in early detection and in delaying the onset of these age-dependent neurodegenerative diseases, respectively. In this review, we highlight how these miRNAs have the potential of being used as disease-based biomarkers (diagnostic tools) and also as therapeutic tools for ameliorating neurodegenerative diseases. However, the field of miRNA-based therapeutics is a developing field in comparison to other oligonucleotide-based therapeutics (siRNA, ASO) and is moving slowly in brain-related disorders in comparison to various forms of cancer. First, it is essential to conduct more detailed basic research to better characterize how miRNAs target molecular and cellular pathways in these neurodegenerative disorders. Second, it is very important to systematically map the on- and off-target toxic effects of the potential miRNAs. Even though these small noncoding RNAs have shown promising results in therapeutically tackling various peripheral diseases, these two factors are an important prerequisite for miRNA’s effective clinical application in AD and other neurodegenerative disorders.

## 2. miRNA Biogenesis Dysregulation in Neurodegeneration

RNA polymerase transcribes primary transcripts (pri-miRNA) from miRNA genes. These transcripts are processed in the nucleus by the Drosha enzyme to produce a hairpinlike precursor miRNA (pre-miRNA). Exportin5 transports the pre-miRNA from the nucleus to the cytoplasm. In the cytoplasm, this pre-miRNA is sheared by Dicer to give rise to the mature miRNA. The mature miRNA collectively with the Ago1 and Ago2 (Argonaute-1 and Argonaute-2) proteins containing RISC (RNA induced silencing complex) binds to the 3′UTR of target mRNA leading to post-transcriptional inhibition or degradation. Dysregulation in Dicer, Drosha, and RISC complexes leads to disruption of miRNA biogenesis, defective cellular processes, and the onset of neuronal degeneration. Several in vitro and in vivo models with loss of functional components of the miRNA biogenesis pathway exhibit progressive neuronal loss. Deletion of Dicer in postmitotic midbrain dopamine neurons causes a progressive loss of cells in mouse models, suggesting an essential role of miRNAs in the differentiation and maintenance of dopaminergic neurons. Loss of miR-133b increases dopamine release in cell cultures, suggesting a role of these small miRNAs in dopamine neural function [20]. 

Smith et al. created a forebrain-specific Dicer conditional knockout mouse in which postmitotic neurons were shown to have increased levels of APP isoforms. MiR-124 plays a pivotal role in neuronal maintenance and splicing. Elevated levels of miR-124 in Neuro2a cells induced skipping of exons 7 and 8 by inhibiting PTBP1 (Poly-pyrimidine tract binding protein 1). Supporting the above finding, miR-124 levels were also found to be lowered in the brains of AD patients [21,22,23]. The essential role of Dicer in CNS was also reflected in the loss of Dicer-1. Dicer-1 mutation dramatically increased neurodegeneration mediated by the truncated form of Ataxin-3 (a mutant form of the spinal cerebellar ataxia type 3) in *Drosophila* and human cells [24]. Genetic ablation of Dicer in Purkinje cells led to cerebellar degeneration and ataxia [25]. Therefore, neurodevelopmental defects that arise as a result of the global loss of Dicer activity need to be interpreted by using the loss of function approaches. This will enable us to understand the functions of specific miRNAs in different aspects of neuronal development and may prove to be more informative. The results of experiments related to removing Dicer support the hypothesis that defects in the miRNA regulatory network in the brain are the potential cause of neurodegenerative disease.

The Drosha complex was found to be associated with TDP-43 (Transactivating response region DNA-binding protein), known to be an important molecular player in both sporadic and familial ALS patients [26]. Mutations in RNA binding protein, FUS/TLS (fused in sarcoma/translocated in liposarcoma) are also found in ALS patients. FUS/TLS protein binds to pre-mRNA molecules and determines their fate by regulating splicing, transport, stability, and translation. It has been shown that FUS/TLS promotes the biogenesis of specific miRNAs via employing Drosha to primary miRNA transcripts [27]. Downregulation of FUS/TLS in neuroblastoma cell lines affects the biogenesis of miRNAs and its recruitment at the chromatin (where it directly binds pri-miRNAs), facilitating Drosha loading. In the miRNA biogenesis process, components of RISC participate in neurodegeneration. Investigations in flies have demonstrated that the mutated form of LRRK2 (leucine-rich repeat kinase2) which is closely associated with PD, is responsible for reduced miRNA-mediated gene repression. Mutant LRKK2 physically interacts with Ago1 and Ago2, inducing their downregulation in aged fruit flies [28]. In HD the mutant Htt (Huntington) gene inhibits the formation of p-bodies (processing bodies) by interacting with Ago1 and Ago2, which are involved in miRNA biogenesis [29]. This hints toward a possibility of miRNA dysregulation in the brains of HD patients.

## 3. Tools for Detecting miRNA Targets

Computational tools have allowed researchers to locate the possible targets of miRNAs and predict miRNA:mRNA interactions in a fast and efficient way. This way, miRNAs can be researched more closely in relation to their involvement with various diseases through their numerous mRNA targets. Some popular prediction tools being used today are TargetScan, miRanda, RNA22, miRBase, PicTar, and PITA to aid in figuring out how the miRNA is acting on the mRNA. These computational tools are based on algorithms that help in locating possible or putative miRNA:mRNA interactions in a miRNA sequence database [30,31].

TargetScan [31,32,33] was the first algorithm to aid in predicting miRNA targets, and has many properties such as Pct scores, context + scores, and TargetScanS that made the predictions much more accurate. When the search has been loaded for a specific miRNA for its potential gene targets across various organisms such as humans, mice, zebrafish, *Drosophila*, etc., the analytical scan will range from highest to lowest probability. The probability is determined by the Pct score, conserved regions, and binding site for all possible miRNA targets analyzed by the algorithm. The miRanda web-based tool is another known algorithm to help identify target predictions of miRNA:mRNA interactions. This algorithm can aid in classifying genes that act in the endogenous miRNA gene regulation system [31,34]. RNA22 uses the patterns found in the sequence in its algorithm to locate potential miRNA target sites and the heteroduplex according to the target site [35]. Another popular algorithm tool used to detect miRNA targets is MiRBase. MiRBase is an extensively large online database that contains information regarding the nomenclature, sequence data, target sites, and annotation that serves as a useful resource for those in the bioinformatics field. The data is organized in a group of miRNA sequences in the related genome. These groups can be picked up in the entry search due to the overlap of miRNA sequences that consist of annotated transcripts that include both protein and noncoding transcripts [36]. PicTar [37] is a tool primarily used to aid in identifying miRNA target site predictions for the human genome as well as the mouse genome. PITA’s (probability of interaction by target accessibility) procedure of identifying miRNA targets is distinct by scanning the UTR (untranslated region) of interest for possible miRNA targets via its seed matching tool, eventually scoring the potential site [37,38]. 

There are several aspects that may assist in ease for computational analysis surrounding miRNAs such as the 3′ UTR sequence, seed region, the relationship between free energy and the interaction, and the miRNA 3′ UTR binding site. The 3′ UTR sequence of the mRNA strand is used to aid in discovering any possible miRNA:mRNA interactions by performing an unbiased algorithm-based search in finding matches to all the annotated 3′ UTR target sites of the selected organism. Messenger RNAs are easily used as a target and can be identified through the database. Many miRNAs have conserved sequences in their 3′ UTR recognition sites that possess a high percentage of nucleotide sequence identity across various organisms. A seed region is considered a conserved piece of the sequence that aids in identifying miRNAs binding to similar targets across different organisms spanning different taxonomical classes and species. As the amount of free energy (kcal/mol) is closer to the minimal level, it makes it easier to test the binding strength between miRNA and the putative target mRNA. The 3′ site of the miRNA’s sequence is located at the 3′ end with the Watson–Crick base pairs between nucleotides 13 and 16, which in turn could help signal the power of gene downregulation. These parameters help ease the number of false possibilities from the search in the algorithms. The scores of each tool applied to predict sequence-based miRNA targets are calculated differently.

Each tool has its own distinct way of making these predictions, and an analysis used with any of the prediction algorithms mentioned above will provide results that may or may not be identical. However, these tools do provide a great way to reduce the number of target predictions. It is important to select the tools that predict miRNA-mRNA interactions in an accurate manner and with lower false positives. Therefore, it is essential to understand the biological aspects used by each prediction tool. Each program has its strengths and weaknesses, and the researcher needs to choose the appropriate tool depending on the requirements. Table 1 summarizes the characteristics, advantages, and disadvantages of a few of these tools. The tool chosen should be able to identify the accurate target and eliminate false positives and incorporate false negatives in the analysis. It is recommended to use more than one tool to increase the choice of candidates with the greatest likelihood of being experimentally validated.

## 4. miRNAs in Alzheimer’s Disease

AD affects about 60% of the elderly population. It is associated with loss in neuronal tissue, memory loss, impaired cognitive functioning, and impaired learning. The exact cause of AD is still unknown. Previous research shows that there are two biomarkers strongly associated with AD. The first biomarker, tau, a microtubule-associated protein promotes vesicle transportation. In AD, hyperphosphorylation of tau causes it to lose its affinity to other molecules. Consequently, this hyperphosphorylated tau develops a stronger affinity for other tau molecules. This causes them to stick together to form Tau aggregates [39]. Elevated levels of tau aggregates lead to a decrease in neuronal communication. The second biomarker is amyloid-β, which is a product of the APP (amyloid-β precursor protein). This molecule is known to form amyloid-β plaques in AD patients [40]. Recent studies show that there is another biomarker independent of tau hyper-phosphorylation and amyloid-β plaques, known as miRNAs. Research investigations over the years discuss the presence of miRNAs circulating within the blood and cerebrospinal fluid.

Some of the miRNAs consistently identified in AD are: mir-9, mir-181, and mir-29, all of which play a role in inflammation and immune response [41]. A study by Pan et al. [40], showed that miRNAs regulate amyloid-β levels. Higher levels of amyloid-β are associated with amyloid-β plaques. Additionally, they found that mir-15 plays a role in the rate of neuronal apoptosis by regulating levels of BCL2 (BCL2 apoptosis regulator). BCL2 is a caspase protein but mir-34 inhibits BCL2 translation. In this study, mir-128 was shown to play a role in tau aggregation and degradation by regulating the synthesis of BAG2 (BAG cochaperone 2), a cochaperone protein that also plays a role in neuronal apoptosis. Researchers also observed that mir-124, mir-132, and mir-9 change the levels of au accumulation in AD [40]. In a study about the role of miRNA in tau metabolism, the mir-15 family was shown to play a role in tau phosphorylation. In a transgenic mutant mouse model, there was an increase in tau when ERK phosphorylation was present in AD. The ERK gene family plays a role in regulating neuronal apoptosis and brain development. Therefore, mir-15 regulates tau phosphorylation by regulating ERK1/MAPK3 (mitogen-activated protein kinase 3) or ERK2/MAPK1 (mitogen-activated protein kinase 1). 

Mir-132 also regulates tau through alternative splicing. Mir-132 regulates PTBP2 (polypyrimidine tract binding protein 2) which is a protein involved in neuronal splicing and regulates tau exon 10 splicing [39]. In another study about tau phosphorylation, researchers investigated whether there is a correlation between miRNA dysregulation and neurodegeneration. Their results showed that miRNA dysregulation does play a role in neurodegeneration by decreasing mir-219 expression. Mir-219 modulates tau toxicity in vivo and regulates tau expression at a posttranscriptional level. In a *Drosophila* model, researchers found that an increase in mir-219 leads to a decrease in the levels of tau protein while a decrease in mir-219 increases it [42]. Hernandez-Rapp et al. performed another study investigating three miRNAs associated with memory deficits in neurodegenerative diseases: mir-132, mir-124, and mir-34. Mir-132 regulates genes associated with neuronal plasticity, growth, and survival. Mir-132 is usually found localized at a synapse and regulates synaptic proteins. In mouse models, high levels of mir-132 are associated with increased learning and memory; however, overexpression of mir-132 can lead to a decrease in long-term potentiation and impaired short-term recognition memory in AD. Hernandez-Rapp et al. found that an increased amount of mir-132 leads to an increased expression of FOXO3a (forkhead box O3a) in AD which is associated with higher rates of neuronal apoptosis. Conversely, researchers also observed that increased levels of FOXO3a are associated with decreased levels of mir-132 in the temporal lobe. This indicates that the expressed levels of FOXO3a are not parallel to mir-132 levels in the temporal lobe [43]. 

Researchers observed that mir-124 conserves synaptic plasticity and preserves neuronal identity. The direct interactors of mir-124 are Egr1 (early growth response 1) and AMPAR GluA2 (α-amino-3-hydroxy-5-methyl-4-isoxazole propionic acid receptor or AMPA receptor, GluA2 subunit). Egr1 plays a role in stabilizing synaptic plasticity while AMPAR GluA2 plays a role in neuronal activity and cognitive function. Mir-34 is a gene family that consists of three domains: mir-34a, mir-34b, and mir-34c. The dysregulation of this gene family is commonly found in neurodegenerative disorders. Research suggests that mir-34c may play a role in neuronal signaling by regulating AMPAR and mGLUR7 (metabotropic glutamate receptor 7), a cytoskeleton protein. In wild-type mouse models, mir-34c played a role in memory impairment and neurodegeneration. Direct mir-34c interactors include: VAMP2 (vesicle associated membrane protein 2), SIRT1 (sirtuin 1), and Notch1 (notch receptor 1) [43].

In a study exploring the advantages of using miRNAs as a therapeutic target for Alzheimer’s disease, researchers found a group of miRNAs that regulate BACE-1 (β Secretase 1) in different ways. These miRNAs also regulate BACE-1 to either increase or decrease amyloidosis. In this study, researchers found that restoring downregulated levels of certain miRNAs would decrease levels of BACE-1 in AD patients. The study proposed inserting a synthetic miRNA, which is comprised of a strandlike guide mRNA and another strand linked to a molecule such as cholesterol. The first strand is referred to as the “antisense” strand, which prevents the synthetic miRNA from binding to a target mRNA. The second strand linked to a molecule increases cellular uptake once it binds to RISC which will then restore the functions of formerly downregulated miRNA [44]. The various miRNAs and their molecular target effectors which are known to cause or modulate AD are depicted in Figure 1 [39,40,43,44,45]. Additionally, we have tried to summarize some other miRNAs modulating AD in a tabular form (Table 2) to highlight all the information that is available regarding how these miRNAs levels are altered in AD and the potential mechanism underlying their regulation of effector targets in the disease pathogenesis [42,46,47,48,49,50,51,52,53,54,55,56,57,58,59,60,61,62,63,64,65,66,67,68,69].

## 5. miRNAs in Other Neurodegenerative Diseases

ALS is a neurodegenerative disorder that occurs with a worldwide incidence of approximately 1.7–2.3 cases out of 100,000 people per year, usually among people in their 50s and 60s [70]. Premature death occurs within 3–5 years of clinical onset from degeneration of specific motor neurons that spread, leading to muscle weakness and eventually muscular atrophy [70,71]. Genes and environmental factors contribute to the etiology of ALS; however, hereditary variables are followed by only 10% of diagnosed cases [72]. The most common genes associated with ALS include *TDP-43* (TAR DNA-binding protein), *FUS* (fused in sarcoma) [73], *NEFH* (ALS2 neurofilament heavy peptide) [72], *C9ORF72* (Chromosome 9 open reading frame 72) [74,75], and Cu/Zn Super Oxide Dismutase 1 (*SOD1*) [76]. Nonetheless, the mechanisms of the genes and their association with the pathogenesis of ALS are currently being studied [77].

MiR-206 is a skeletal muscle-specific miRNA and was found to be upregulated during reinnervation of the sciatic nerve in SOD1 transgenic mice [78]. The dysregulation of the miR-206 exacerbated the progression of ALS and lessened the lifespan of *SOD1* transgenic mice. It was found that miR-206 inhibits the translation of *HDAC4* (histone deacetylase 4) which activates the secretion of FGFBP1 (fibroblast growth factor binding protein 1) and increased miR-206 expression levels to stimulate reinnervation at the neuromuscular junction [78]. From there, the FGFBP1 binds to FGF to induce synaptogenesis [79]. Nevertheless, miR-206 is not the only miRNA involved. It was found that miR-23a, miR-29b, and miR-455 are upregulated in skeletal muscle tissues of ALS patients. However, further research is required to understand their roles in regulating the expression of the mitochondrial genes and causing ALS [80]. 

Another gene associated with ALS is *TDP-43* (TAR DNA-binding protein-43), which is also involved in the miRNA pathway of ALS. TDP-43 is a RNA binding protein that is part of the Dicer and Drosha complex that interacts with the miRNA processing enzyme called Drosha and binds with primary miRNAs to induce the process of miRNA biogenesis [81,82]. It was found that mutations in *TDP-43*, *FUS*, *SOD1*, and other genes associated with ALS induced phosphorylation of eIF2 (an important translation initiation factor) and stress granule formation, leading to activated stress response [83]. The activated stress response altered the shape of the Dicer complex, simultaneously downregulating miRNA expression and diminishing the pre-miRNA processing, eventually leading to motor neuron degeneration. Although identification of miRNAs that influence TDP-43 is currently under investigation, mutations in TDP-43 result in distinct expressions of miRNAs, especially miR-132, miR-143, and miR-558, that contribute to ALS pathogenesis [84]. Additionally, among ALS patients with *TDP-43* mutation, upregulated miR-9 expression levels were recognized [85]. Moreover, another gene associated with ALS is *FUS*, and *FUS/TLS* was found to be associated with ALS [86]. *FUS/TLS* is a RNA protein that binds to DNA and RNA, participates in a variety of cellular processes by regulating splicing, transcription, translation, transport, and stability [87,88]. *FUS/TLS* incorporates primary miRNA transcripts with Drosha to stimulate the synthesis of specific groups of miRNAs such as miR-132, miR-134, and miR-9 that are involved in neurogenesis and neuroplasticity [89]. MiR-134 is also involved in neural development and dendritogenesis [90]. The miRNA, miR-132, modulates its targets acetylcholinesterase and tau to regulate the structure and growth of neurons [23,91,92,93]. On the other hand, miR-9 is involved in regulating *MAP1B* (microtubule-associated protein 1b) mRNA translation to govern axon growth [94]. Research has suggested that the ALS phenotype in patients may have been caused by mutations in *FUS/TLS* which can influence the function of miRNAs.

HD is a neurodegenerative disease caused by a CAG repeat expansion of the Huntington gene (*Htt*). The mutation of the *Htt* gene leads to progressive loss of neurons in the striatum and the cortex, resulting in loss of cognitive and motor control function in HD patients. The *Htt* gene was found to interact with REST (transcriptional repressor RE1-silencing transcription factor), also known as NRSF (neuron-restrictive silencing factor), which is known to inhibit genes responsible for neuronal differentiation [95]. Qiu et al. [96] stated that the miRNA activity has been affiliated to HD on the account of association among a mutated *Htt* with alterations in miRNA mechanisms by its involvement in Ago2 and P-bodies, cytoplasmic sites of RNA metabolism, RNA interference, and miRNA processes [29,97,98]. Likewise, the association between miRNAs and HD was even further supported by studies on transcriptional pathways. Pathways, involving REST are the most well-known among all [99,100]. The REST transcription factor interacts with the mutant *Htt*, contributing to the development of HD. Ballas et al. showed that REST regulates the transitions from pluripotent to neural stem/progenitor cell and from progenitor cell to mature neuron. During the transition to progenitor cell, REST is degraded to extremely low levels sufficient to maintain neuronal genes in an inactive state. When progenitors differentiate into neurons, REST and its corepressors dissociate from the RE1 site, initiating activation of neuronal genes [101]. REST is expressed in the cytoplasm at low levels and binds with the *Htt*. Mutated forms of the *Htt* lack the ability to bind to REST, resulting in its movement from cytoplasm to the nucleus [95]. Consequently, REST inactivates genes that promote neuronal regulation by binding to RE1 in the nucleus which recruits corepressors mSin3 and MeCP2 (methyl CpG binding protein 2) [102].

REST is known to have numerous targets, and one of the main targets of REST is the trophic factor called the brain-derived neurotrophic factor (BDNF). Johnson et al. wrote that BDNF is transported and secreted onto the striatal neurons in the cerebral cortex and that the absence of BDNF induces death of striatal neurons, demonstrating the importance of BDNF in neuronal survival. REST’s targets miR-9, miR-29a, miR-29b, miR-124a, miR-132, miR-135b, miR-139, miR-203, miR-204, miR-212, miR-330, and miR-346, were downregulated in the cortex of a R6/2 mouse, a transgenic animal model of HD [103]. However, only miR-132 regulated by CREB (cAMP response element-binding protein) and BDNF pathway, was confirmed to be downregulated in parietal cortical tissue of humans [103,104,105]. Downregulation of miR-132 in HD may affect the mechanisms of the BDNF and neurogenesis due to miR-132′s targets MeCP2, which play a role in the feedback loop of BDNF expression in neurons, and p250GAP (brain-enriched NMDA receptor-interacting RhoGAP), which regulates neuronal morphogenesis [106,107]. Another well-known target of REST is miR-9. MiR-9-5p and miR-9-3p exhibit decreased expression in the early onset of HD. It was found that miR-9 induces neuronal morphogenesis by repressing REST expression and BAF53a (ACTL6A or actin-like 6A), which inhibits neurogenesis by regulating chromatin remodeling [108]. Furthermore, it was stated that miR-9′s role in neurogenesis is supported by phosphorylation of STAT3 (signal transducer and activator of transcription 3) and regulation of its protein targets in the proteome, including the BDNF [109,110]. Additionally, in mouse models miR-9 plays a role in the differentiation of Cajal-Retzius cells found in the hippocampus and neocortex [111,112]. Nevertheless, REST and CoREST, a corepressor for REST, are also predicted targets of miR-9-5p and miR-9-3p, indicating the possibility of a double-negative feedback loop between miR-9 and REST [112]. In essence, the downregulation of miR-9 may influence the mechanisms of neurogenesis and perpetuate the development of HD. 

Other miRNAs that contribute to HD include miR-22 and miR-128. MiR-22 and miR-128 are decreased in HD brains, while miR-128 was proven to exhibit decreased expression in the brains of mice, monkeys, and humans [105,112,113,114]. MiR-22 and miR-128 are important for neurogenesis and neuronal survival, especially the miR-22 targets of histone deacetylase 4 (HDAC4), REST corepressor 1 (Rcor1), and regulator of G-protein signaling 2(Rgs2), which were demonstrated by luciferase assay [115]. Furthermore, it was demonstrated that expression of miR-22 inhibits proapoptotic proteins, reduces the activity of caspases, and protects neurons from stress, which may illustrate miR-22′s involvement in the pathogenesis of HD by contributing to neurogenesis and neuronal survival [115]. On the other hand, miR-128 targets the HTT interacting protein 1 (HIP1), SP-1, HTT, and GRM5, to regulate the HTT signaling pathway which perhaps explains the role of miR-128 in the pathogenesis of HD [114]. There are other miRNAs that contribute to the development of HD. MiR-17-3p, miR-128, miR-139-3p, miR-196a, miR-222, miR-382, miR-433, miR-483-3p, miR-485-5p, miR-486, and miR-500 were found to be downregulated in HD brains [112,113]. On the other hand, miR-100, miR-151-3p, miR-16, miR-219-2-3p, miR-27b, miR-451, and miR-92a were found to be upregulated in HD tissues [113]. Nonetheless, one must consider that the miRNAs attributed to Huntington’s disease may also be associated with neurodegenerative diseases as a whole. The miRNAs listed in this review may not be a complete list and more research needs to be conducted to study the association of miRNA and Huntington’s disease and decipher novel miRNAs involved in the disease pathogenesis.

Parkinson’s Disease (PD) is a neurodegenerative disorder characterized by progressive neuronal degeneration, mainly in the substantia nigra [116] with symptoms of bradykinesia, rigidity, resting tremor, and posture instability [117]. There are numerous genes associated with PD including Parkin (PARK2), PINK1 (PTEN induced kinase 1, PARK6), DJ-1 (protein deglycase, PARK7), LRRK2 (leucine-rich repeat kinase 2, PARK8), and ATP13A2 (ATPase cation transporting 13A2, PARK9) [118]. One of the prominent pathological hallmarks of PD development is the accumulation of SNCA (synuclein α) in the Lewy bodies which then binds to the ubiquitin in the cells [118,119,120]. The specific functions of SNCA are unknown, however, miR-7 and miR-153 were found to restrain SNCA expression levels [121,122]. Research investigations found that miR-7 and miR-153 regulate SNCA by posttranscriptionally binding directly to the 3′ UTR of the SNCA gene, reducing the expression levels of SNCA [122]. On the other hand, Junn et al. found that miR-7 inhibited SNCA expression in human neuroblastoma cells by oxidative stress pathways [121]. Hence, it was proposed that oxidative stress may be involved in PD pathogenesis by miR-7 downregulation. Nonetheless, there are other factors that contribute to PD. Another prominent contributor to the pathogenesis of PD that is also involved with dopamine-producing cells, especially among sporadic PD patients, is LRRK2. LRRK2 is involved in the association of reduced dopamine levels and PD by negatively controlling Let-7 and miR-184 in dopamine-producing cells [28]. It was discovered that miR-181, miR-19, and miR-410 binding sites were found in the 3′ UTR binding site of LRRK2 but did not demonstrate significantly altered levels of expression in PD patients [123]. On the other hand, miR-205, which also targets 3′ UTR of LRRK2, was downregulated in the frontal cortex and striatum of sporadic PD patients. It was found that the repressed miR-205 induced overexpression of LRRK2 and elevated levels of LRRK2 were detected in sporadic PD patients. Consequently, upregulation of miR-205 indicated inhibition of the LRRK2 expression and ultimately hindered the growth of neurites [28,123,124]. Although the exact mechanisms of miR-205 levels and LRRK2 expression changes are unknown, Cho et al. [123] suggested a consideration in miR-205 and apoptosis resistance to understand how elevated levels of miR-205 inhibit LRRK2 expression. Bhatnagar, Li, and Padi et al. [125] stated that DNA methylation may influence the expression of miR-205. BCL2L2 (BLCL2 like 2 gene encoding Bcl-w) is an antiapoptotic gene and a target of miR-205. The introduction of DNA methylation in a prostate cancer cell led to an increase in miR-205 expression levels. This facilitated the response of prostate cancer cells to apoptosis from chemotherapy [125]. Hence, miR-205 may have an important role in making the cells responsive to apoptosis, and there is a possibility that the miR-205 may be involved in neuronal survival. In turn, the inhibition of miR-205 may contribute to the upregulation of LRRK2, contributing to the development of PD. Additionally, degeneration of dopaminergic cells is commonly portrayed in PD patients.

Kim et al. mentioned miRNAs that are associated with lower levels of dopamine and that one of the miRNAs is miR-133b, which was found to be downregulated in sporadic PD patients [20]. According to Kim et al., downregulation of miR-133b in embryonic stem cells amplifies the release of dopamine due to increased terminal differentiation of dopaminergic neurons. On the other hand, upregulation of miR-133b contributes to the release of dopamine by regulating Pitx3 (pituitary homeobox 3 transcription factor (Pitx3) in a feedback loop) [20]. Pitx3 is responsible for the development of dopaminergic neuronal differentiation when overexpressed by miR-133b. The increased expression of miR-133b results in decreased expression of the dopamine active transporter (DAT) and tyrosine hydroxylase (TH), and inhibition of terminal differentiation which may decrease the release of dopamine. In essence, miR-133b may be involved in PD development by participating in the differentiation of dopaminergic neurons. Another miRNA involved with dopaminergic neurons is miR-433. Itoh et al. stated that SNP (single nucleotide polymorphism) in the 3′ noncoding region of FGF20 can be a risk factor for Parkinson’s disease. They reported that the risk allele disrupts a binding site for miRNA-433, increasing FGF20 mRNA translation, and elevated levels of FGF20 mRNA translation has been correlated with increased α-Synuclein expression [126]. However, Wang et al. indicated that not all SNPs are susceptible to inducing PD pathogenesis. MiR-433 may prevent the impairment of SNP rs12720208 by FGF20, but more research is required to learn the association among the FGF20 gene, miR-433, and PD [127]. Moreover, miR-124a is also involved in dopamine-producing cells and PD by regulating FoxA2 (forkhead box A2), a transcription activator that plays a crucial role in dopamine cell production in the midbrain of humans and rodents [128,129]. FoxA2 was involved in regulating glucose metabolism and insulin secretion. ATP-sensitive K+ channel activity in neuronal homeostasis in genes demonstrated the importance of miR-124a in dopaminergic neuronal survival. 

Other miRNAs that contribute to PD include miR-107 and miR-34. MiR-107 was found to be downregulated in the midbrain of PD patients [20]. P53 regulates the expression levels of miR-107 by binding to the 5′ UTR of the miR-107’s parent gene, PANK1 (pantothenate kinase enzyme 1) [130]. Additionally, miR-107 has two targets—progranulin and CDK6 (cyclin dependent kinase 6). Progranulin is a growth factor that was found to contribute to the development of frontal temporal dementia [131,132]. On the other hand, miR-107’s target CDK6 regulates the cell cycle [133,134]. Inhibition of miR-107 may result in upregulation of CDK6 and facilitate cell cycle re-entry, leading to the death of mature neurons [133,134]. Hence, miR-107 may influence PD pathogenesis through the exposure of neurons to cell cycle re-entry. Furthermore, miR-34 was found to be downregulated early in PD patients [135]. MiR-34’s targets, CDK4 (cyclin dependent kinase 4) and cyclin D2 are cell cycle regulators and are involved in the death of postmeiotic neurons [136]. Additionally, p53 (tumor protein P53) not only regulates miR-107 but also miR-34a by binding to the promoter of miR-34a [137]. In turn, miR-34a inhibits the expression of HDM4 (human MDM4), which consequently reduces the expression of p53, creating a feedback loop between p53 and miR-34a [138]. MiR-34 is also involved in cell survival by disturbing the mitochondrial membrane and elevating oxidative stress levels when miR-34 is downregulated in SH-SY5Y cells [135]. In summary, miRNAs are important regulators of PD pathogenesis, and more research needs to be conducted to understand the detailed mechanism underlying this regulation (Figure 2).

## 6. miRNAs as Biomarkers and Therapeutics

miRNA in biomarker study: Biomarkers are genetic, molecular, or biochemical components that facilitate the identification and analysis of pathological processes when expressed in the human body [141,142]. They can be used as an apparatus to assess the different stages of diseases, especially of the early or preclinical stages. The study of biomarkers in the early stage of any disease may enable patients to receive early treatment and help in the development of therapeutic strategies. Subsequently, when biomarkers are identified at early stages, they can be used as tools to examine the response of treatment to the rate of disease progression. miRNA has been advocated as a possible biomarker for different types of diseases in regard to diagnosis and treatment. The range of diseases spans from neurodegenerative diseases such as AD [143] to different types of cancer, including gastric cancer, colorectal cancer, lung carcinoma, oesophageal cancer, and breast cancer [144,145,146,147,148]. An additional example of a miRNA biomarker study is the utilization of their levels in cerebrospinal fluid (CSF) as minimally invasive detection for central nervous system diseases such as lymphomas and gliomas [149,150]. In essence, although miRNAs are not direct causes and indicators of diseases, the involvement of miRNAs in the pathogenesis of diseases provides the beneficial utility of miRNAs as potential biomarkers. 

Utilization of circulating miRNA in body fluids correlating with the statistically significant altered miRNA expression levels in AD was widely accepted as an important diagnostic parameter in miRNA biomarker research. As stated in the article written by Schipper et al., they were the first to conduct miRNA biomarker research for AD and claimed that the microarray analysis found upregulation of miRNA levels in Alzheimer peripheral blood mononuclear cells (PBMCs) [151]. On the other hand, Leidinger and Backes et al. were the first to use sequencing analysis to determine the accuracy of difference in miRNA levels between control and AD patients. Based on their research, miR-12 indicated 92% accuracy, 95% specificity, and 92% sensitivity in the differential levels between control and AD patients [9]. Adding on, Tan and Yu et al. demonstrated the use of RNA sequencing and qRT-PCR of serum samples to detect the difference in miRNA levels expressed. MiR-98-5p, miR-885-5p, miR-483-3p, miR-342-3p, miR-191-5p and miR-let-7d-5p demonstrated differential expression levels between control and AD patients [152]. MiR-342-3p demonstrated the highest sensitivity in the differential expression by 81.5% and specificity by 70.1% [152]. On the other hand, another study, conducted by Tan et al. revealed that miR-125b and miR-181c was downregulated while miR-9 was upregulated in the serum of AD patients [10]. Additionally, several research works showed that miRNA biomarkers were detected in the plasma samples for MCI (mild cognitive impairment) [141,142,153,154]. Sheinerman et al. indicated that the difference in expression of miR-132 related miRNAs (miR-128/miR-491-5p, miR-132/miR-491-5p, and miR-874/miR-491-5p) and miR-134 related miRNAs (miR-134/miR-370, miR-323-3p/miR-370, and miR-382/miR-370) displayed a sensitivity of 79–100% and specificity of 79–95% between controls and MCI patients [153]. Moreover, Kumar et al. used nanostring technology in plasma samples and the discovered seven miRNAs (let-7d-5p, let-7g-5p, miR-15b-5p, miR-142-3p, miR-191-5p, miR-301a-3p, and miR-545-3p) showed 95% accuracy of discernment between AD patients and controls [155]. In fact, Lehmann et al. also showed upregulation of let-7b, a miRNA that activates Toll-like receptor 7, in CSF instead of serum samples [156].

Other than plasma samples, studies using CSF and serum samples also depicted differences in miRNA expressions between AD and control. Sala et al. utilized the application of qRT-PCR on CSF samples to show decreased levels of miR-27a-3p in AD patients [157]. Additionally, in a study conducted by Bekris et al. and Burgos et al., microarray and qRT-PCR were used on plasma and CSF samples. The results depicted a positive correlation between the neuritic plaque score found in the plasma and upregulation of miR-15a in the hippocampus of AD patients [158,159]. Furthermore, Burgos and Malenica et al. demonstrated association of Braak stage, dementia status, plaque, tangle densities, and Lewy body pathology with altered miRNA levels in CSF and serum [158]. In addition, Lukiw et al. discovered that miR-146a and miR-155 were found in CSF sample of AD while Kiko et al. indicated that miR-29a and miR-29b levels were higher and miR- 34a, miR-125b, and miR-146a levels were lowered in the CSF samples of AD patients. Additionally, miR-34a and miR-146a levels were shown to be downregulated in plasma samples [48,160]. Moreover, Geekiyanage et al. demonstrated that the posttranscriptional regulation of miR-137, miR-181c, miR-9, and miR-29a/b is used as a mechanism by serine palmitoyl transferase (SPT) to regulate amyloid-β (Aβ) levels in CSF samples [161]. Nevertheless, Muller et al. demonstrated that only miR-29a increased by a factor of 2.2 in CSF samples of AD patients among the various miRNAs being studied [162]. The application of CSF samples provides a possibility of detecting circulating miRNAs for diseases.

Despite the ongoing studies, further research is needed for implementing miRNA as biomarkers for AD. Sorensen et al. conducted a study analyzing the levels of miRNAs in CSF and blood of AD patients. It was discovered that 168 miRNAs were found in the blood, while 52 miRNAs were found in CSF samples [163]. The application of CSF to detect miRNAs was highly recommended in comparison to blood. Blood samples are easily obtained at low-cost and low-risk. However, there is a chance of additional factors that contribute to what is found in the blood other than from the brain due to blood circulation. On the other hand, CSF may be a more reliable biomarker because it is protected by the blood-brain barrier which decreases compounding variables that may influence the composition of miRNAs in AD. In fact, miR-29c- 3p, miR15a-5p, and let-7i-5p were detected in CSF samples and indicate the possibility of association with APP and BACE1 [163]. Wu et al. conducted a meta-analysis of miRNA biomarker research studies and wrote of the need for independent cohort studies for miRNAs as biomarkers for MCI and AD. The analysis demonstrated that miR-29b, miR-181c, miR-15b, miR-146a, and miR-107 showed consistent differential expression in at least two independent studies while mir29b, miR181c, and miR15b from CSF samples were shown to have a significant association with AD [164]. On the other hand, miR-132 and miR-107 showed a consistent association with MCI, and most miRNAs had greater than 0.75 in sensitivity and specificity to differential miRNA expression levels [164].

Few studies exhibited the need for standardized methodology and miRNA measurement. When regarding the use of circulating miRNAs as biomarkers, one must consider the external and internal factors that may influence miRNA levels, ranging from genetic variation, race, gender, inflammatory status, and lifestyle factors to differences in methodology, such as the techniques used to process samples and measure the miRNA levels [164,165]. The concentrations of miRNA levels vary in serum and plasma samples within the same person, and an independent research study showed that different concentrations and types of miRNA were found in blood, plasma, serum, or even exosome samples [166]. In addition to various factors that influence miRNA levels, the process of analyzing data also affects the statistical significance of their levels in biomarker studies. Some researchers selected miRNAs with at least a 2-fold change between AD patients and controls while others selected miRNAs with at least a 1.5-fold difference between MCI/AD patients and controls [152,155,164]. The variation of miRNA measurement requires a more standardized protocol, such as consistent sample preparation, statistical calculations, and systematic analysis of miRNA levels that are unvarying. 

Although the need for a standardized protocol is addressed, one must recognize that miRNA biomarker research is an emerging field that will undergo constant changes and discoveries. A difference in miRNA expression levels may provide an understanding of the mechanisms of AD. Several miRNAs expressed different results among different studies, depending on the biological source of the samples used for miRNA profiling. For example, miR-9 was shown to be downregulated or upregulated in the blood, CSF, and the brain, and such different results may have influenced expression levels of different targets of miR-9 [10,11,158,160,167,168,169,170,171,172]. Oxidative pathways and AD inflammation may induce upregulation of miR-9 while downregulation of miR-9 may increase the expression of BACE1 and Aβ42 [171,173]. In essence, miRNA biomarker research provides expansive potential in learning the pathogenesis of AD and the intricate mechanisms involved that may help us monitor and understand the disease.

miRNAs as therapeutics: It is very challenging to deliver miRNAs into the central nervous system (CNS) because the blood-brain barrier (BBB) prevents the accumulation of active compounds in the CNS, limiting their transfection efficiency. To increase the transfection efficiency of these miRNAs and aid in their blood-brain barrier crossing, two strategies have been formulated. The first one is the restoration of the suppressed miRNA level by miRNA mimics (agonist). The other is inhibiting miRNA function by using anti-miR (antagonist) to repress overactive miRNA function [174,175,176,177,178,179,180]. An overview of the available therapeutic strategies for reinstatement or inhibition of miRNAs is presented in Table 3.

The therapeutic application of miRNAs-based agents is promising due to the following properties: potency, effectiveness, optimum duration of gene expression silencing, simplicity, safety, and easier manufacturing methods. However, the success of RNA therapeutics development for the treatment of neurodegenerative disease is limited owing to several obstructions and challenges, including extracellular and intracellular blockades. The extracellular barriers are responsible for low RNA bioavailability, enzymatic degradation by bloodstream nucleases, rapid renal clearance, phagocytosis by monocytes and macrophages, opsonization by blood complements (e.g., lipoproteins, immunoglobulins, erythrocytes, and serum proteins), and toxicity due to immune stimulation, off-target effects, and diffusion through the cellular matrix. On the other hand, the intracellular barriers include nonspecific targeting to the physiological sites (organs, tissues, or cells), inefficient cellular uptake, and intracellular processing of endosome-targeted RNAs (poor escape from the endosome, inadequate vector unpackaging, and processing by the RNAi machinery). These hurdles need to be overcome before the therapeutic product arrives at the cytoplasm, allowing an improvement on miRNA pharmacokinetic and pharmacodynamic properties [181,182].

RNAi-based therapeutics often have a problem of poor stability because naked nucleic acids are degraded by enzymes before reaching the target sites. Therefore, an effective, safe, and stable biologically responsive delivery system is necessary to protect the nucleic acids from serum degradation and assist their entry into cells [183]. Nonspecific uptake of miRNA-based therapeutic agents by nontarget tissues might lead to potential unwanted off-target effects. To efficiently deliver miRNAs into the body system in a safe and controlled manner remains challenging. Another important factor that controls the uptake and distribution of numerous therapeutics in the CNS is the presence of a large number of specific transporters and receptors, such as carrier-mediated influx transporters (e.g., glucose transporters, large amino acid transporter, monocarboxylate transporters, organic anion transporters, and nucleoside transporters), efflux transporters (called ATP-binding cassette (ABC) transporters that include P-glycoprotein, breast cancer resistance protein, and multidrug resistance-associated proteins), and receptors mediated transporters (such as transferrin receptor, lactoferrin, leptin receptors, insulin receptor, endothelial growth factor receptor), as well as low-density lipoprotein receptors in the brain endothelial cells [178,183]. These receptors facilitate the selective delivery of molecules to the brain, strengthening the BBB (blood-brain barrier) function by effectively removing drugs from the brain and pumping them back into the blood. Because of this BBB transport restriction mechanism, more than 98% of candidate drugs have been abandoned during their development [184,185,186,187,188,189,190,191,192,193,194,195,196].

Within the context of delivery into the brain, the distribution of miRNA-based therapeutics can also be influenced by the route of delivery. To overcome this issue, two methods have been applied for drug transport across the BBB, namely invasive and noninvasive approaches. The invasive approaches include intracerebroventricular infusion, intrathecal, convection-enhanced delivery, and the disruption methods of BBB integrity. These approaches have a high risk of side effects such as infection, edema, and neuron damage and some drawbacks (less safe, inconvenient mode of delivery, and high cost). However, small amounts of drugs are required for the delivery. The noninvasive approaches are better with respect to patient compliance and are more suitable for neurodegenerative disease therapy. These include lipid-mediated drug transport, systemic intravenous administration, intranasal delivery, and strategies using nanosystems. In the past few years, different and efficient carrier systems have been developed and applied in gene therapy trials to promote the transport and delivery of miRNAs-based therapeutics into the brain, increasing the accumulation of these therapeutics in the site of interest, enhancing the silencing potency, thereby making these kinds of RNA therapeutics more effective. In a simple way, delivery methods can be divided into two categories, nonviral (Table 4) and viral systems (Table 5). However, each of these approaches has distinct advantages and disadvantages that require careful consideration (Table 4 and Table 5).

Initially, research focused on the use of viruses as gene carriers because they displayed high efficiency at delivering miRNAs, taking advantage of their efficient cell uptake and intracellular trafficking machinery and enabling long-term gene expression [180,197,198,199] (Table 5). Nonetheless, the research focus over the last few decades has changed to the nonviral transport systems because of the advantages over viral vectors, namely the ability to safely deliver miRNA-based therapeutics in the specific site, reduced cytotoxicity with no or little immune response, high flexibility and easy quality control, relatively high drug loading, low production costs, enhanced intracellular delivery, and early endosomal escape, permitting repeated administrations and inducing sustained expression [180,200,201,202]. However, these systems also present some limitations such as reduced transfection efficiency due to cellular barriers and immune defense mechanisms, poor oral bioavailability, and instability in circulation. There are different types of nonviral delivery systems, such as liposomes, lipoplexes, polymeric nanoparticles, cyclo-dextrins, dendrimers, polymeric micelles, and exosomes, that have been extensively studied for brain drug delivery [194,200,201,202,203,204,205,206,207,208,209,210,211] (Table 4).

The modification of lentivirus tropism has been made toward astrocytes with neuron-specific miR124, to remove residual expression in neuronal cells for cell type-specific gene transfer to the CNS [212]. Effective lentiviral transfection of recombinant human miRNA-7-3 gene into human glioma cells to suppress gliomal cell growth were also reported [213]. Lentivirus-mediated miRNA-210 has been delivered in an ischemic mouse brain and showed improvement of long-term outcomes for stroke therapy [214].

A cationic lipoplexes-mediated carrier system for miR-29b delivery was reported to suppress tumorigenicity by restitution of miR-29b in non-small cell lung cancer. These lipoplexes contained a liposome containing, cationic lipid, 1, 2-di-octadecenyl-3-trimethylammonium propane, a neutral lipid, cholesterol, and d-α-Tocopheryl PEG succinate and was formulated to entrap miR-29b. The positively charged lipoplexes proved to establish interaction with the negatively charged cell membrane, providing efficient cellular uptake to target multiple oncogenes in non-small cell lung cancer cells [212]. Similarly, a cationic liposome vehicle composed of a cationic lipid 2-dioleyloxy-N, N-dimethyl-3-aminopropane, egg phosphatidylcholine, and cholesterol was developed for delivery of miR-122 mimic in hepatocellular carcinoma therapy, resulting in significant inhibition of expression of miR-122 target genes. PEG was attached to the surface of the lipid nanoparticle (LNP) to increase the in vivo stability and circulation half-life time [213]. Targeted delivery of miR125a-5p via the cationic lipid nanoparticle (LNP) platform for the treatment of HER2 positive metastatic breast cancer was investigated. A lipid solution mixture was mechanically extruded to create unilamellar vesicles of LNP, followed by conjugation of hyaluronic acid (HA) on the surface of the LNP. The formulation was further lyophilized and was rehydrated with FITC-Dextran tagged human miR125a-5p mimic solution for entrapping the miRNA [214]. A novel approach that can simultaneously deliver miR-34a and doxorubicin into HA-chitosan NPs (nanoparticles) against triple negative breast cancer was studied. In this approach, anionic HA and cationic chitosan were used to encapsulate negatively charged miR-34a and positively charged doxorubicin through a cross-linker tripolyphosphate [215]. Another study used biocompatible and traceable Poly (lactic acid-co-glycolic acid; PLGA) NPs containing perfluoro-1,5-crown ether that can be tracked by 19F-MRI. Protamine sulfate was then surface conjugated to complex miR-124 to enhance brain repair in PD mice models. The results showed decreased expression of Sox9 and Jagged1, two miR-124 targets and stemness-related genes. The use of miR-124-PLGA NPs demonstrated a new theranostic method for neurodegenerative diseases [186]. Reducible polyethylenimines (PEI) synthesised of high molecular weight PEI (25,000 Da) with cetyl bromide and then conjugated with PLGA polymer and cross-linked with HA facilitated the cellular uptake of tumor-suppressor miR-145 [216,217,218]. A magnetic reagent for efficient transfection (MagRET) comprised of a maghemite core that is surface-treated with lanthanide Ce3/4+ cations was fabricated as a gene carrier. PEI was then attached to this maghemite core to form an antisense miRNAs NP complex for silencing miRNAs [219]. Functionalized carbon nanotubes (CNTs) have been used to regulate target gene expression and angiogenesis. CNTs are coated with polymers, to improve their electrostatic interaction with the negatively charged siRNAs or plasmid DNA, followed by conjugation of miR-503. The results showed increasing nucleic acids loading and improving cell uptake [220].

The oligonucleotide-based therapy in neurodegenerative diseases that entered clinical trials was published in 2014. The delivery platform in these trials was mainly naked ASO delivery without vectors [178]. In summary, all these methods of delivery of miRNAs can be potentially extended as therapeutic interventions for neurodegenerative diseases.

RNA-based therapeutics are emerging as potential drugs in various clinical fields, including neurodegeneration. Even though results using RNA-based therapeutics have been extremely promising, these positive studies do not yet include miRNA-based strategies targeted against AD pathology. The first siRNA drug (patisiran) was approved by the FDA in 2018 for the treatment of hereditary transthyretin-mediated amyloidosis. In this strategy, siRNA was encapsulated in lipid nanoparticles directing it to the liver where it bound to the mRNA of transthyretin and inhibited the production of the mutant protein [221,222,223]. Treatment for spinal muscular atrophy (SMA) was approved in 2016 using a 2′-O-2-methoxyethyl phosphorothioate-modified ASO. The ASO interfered with the splicing of SMN2 mRNA, thereby elevating the amount of functional SMN2 protein, which could compensate for the loss of SMN1 [224,225]. Another RNA-based therapy (zolgensma) was approved by the FDA for SMA, where a functional copy of SMN1 gene is delivered using an AAV9 delivery system [226]. In AD, the most advanced RNA-based therapeutic is based on translational inhibition of *tau* mRNA using an ASO-based strategy (NCT03186989, BIIB080, IONISMAPTRX), and is currently a clinical phase I/II trial study. The miRNA-targeted pharmaceutical arcade is less advanced with numerous ongoing clinical trials (CDR132L, Cardior Pharmaceuticals GmbH; RG012, Genzyme/Sanofi/Regulus Therapeutics; MRG-106, MRG-110, MRG-201, miRagen/Viridian Therapeutics; TargomiRs [227,228,229,230]) with none of them being in AD. However, MRG-107, a miR-155 inhibitor has been pre-clinically validated by miRagen Therapeutics against ALS [231].

Even though various clinical trials to test miRNA-based therapeutics against several peripheral diseases are ongoing, no therapeutic tools have reached clinical trials for the treatment of AD. However, gemfibrozil, a previously FDA-approved drug for decreasing cholesterol and lipids, has undergone a phase I trial to evaluate its ability to increase miR-107 levels for prevention of AD in cognitive health and MCI individuals (NCT02045056). According to reports, 48 control and 24 MCI individuals were treated with gemfibrozil or a placebo. Gemfibrozil was found to be safe, inducing a change in miR-107 plasma levels and a decrease in Aβ42, pTAU, Aβ42/pTau ratio, brain atrophy, and plasma TNFα levels in the treated patients. However, these measures did not reach statistical significance. The field of miRNA-based therapeutics is a developing field in comparison to other oligonucleotide-based therapeutics (siRNA, ASO). First, it is essential to conduct more detailed basic research to better characterize how miRNAs target molecular and cellular pathways. Second, researchers must systematically map the on- and off-target toxic effects. These two factors are important prerequisites for effective clinical application in AD and other neurodegenerative disorders. Improved methods of targeted brain delivery and additional investigation of the tolerability of miRNA restoration strategies are key issues to be resolved. Even though a large number of miRNA-based companies are being acquired by major pharmaceutical companies to help find a novel category of drugs [232], application of miRNA therapeutics in AD is lagging behind other diseases areas such as cancer, which has approximately 30 times more new molecular targets in clinical trials than AD [233]. Technology improvisation, aggressive investment, and research development in the field of neurodegenerative disorders such as AD, are needed to bridge the gap between promising initial miRNA research and clinical application.

## 7. Conclusions

The last decade has seen an increase in the number of groups using in vitro and in vivo models to understand the roles of brain-enriched miRNAs in human neurodegenerative diseases. This paper has provided a comprehensive review of this field, but it is hoped that by highlighting key papers, this review may inspire those new to this area to begin asking new questions and making advancements in the field of miRNAs’ role in causing neurodegeneration and its use in designing effective diagnostics and therapeutic tools. Currently, available data indicates significant regulatory roles of miRNAs in the pathogenesis of neurodegeneration. These remarkable miRNAs could be used as diagnostic markers and therapeutics for many progressive neurodegenerative diseases. However, more research should be performed on the pharmacokinetics of miRNA in the body to understand the threshold copies of miRNA that should be replaced or repressed in each disease state. To design specific miRNA carriers for long-term gene expression and knockdown in CNS is an important challenge for scientists. The utility of all the work performed in this field has aided and will aid in discovering novel pathways and molecular mechanisms underlying severe progressive age-dependent neurodegenerative disease. The work done so far in this field is just incremental steps toward designing more effective miRNA-based diagnostics and therapeutics for the future.

## Figures and Tables

**Figure 1 genes-13-00425-f001:**
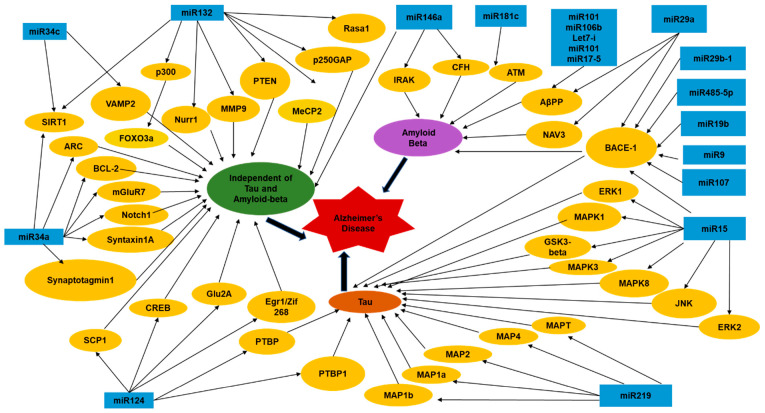
Schematic diagram showing the various miRNAs and their molecular targets involved in regulating AD [39,40,43,44,45]. These miRNAs modulate the intensity of AD by modulating levels or functions of tau and abeta. Additionally, they modify tau and abeta independent cellular pathways leading to neuronal toxicity.

**Figure 2 genes-13-00425-f002:**
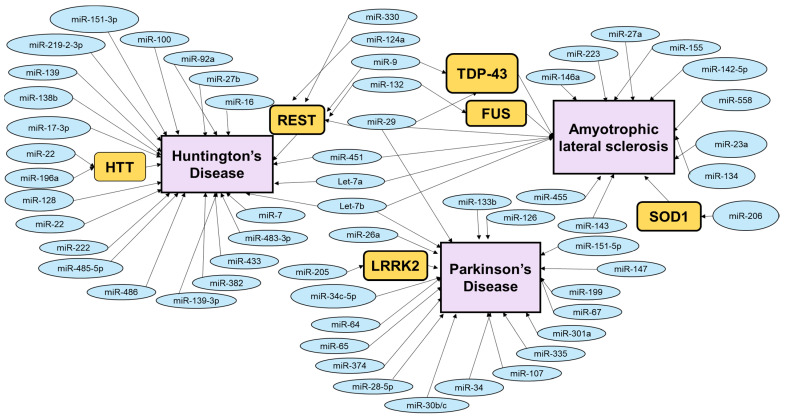
Schematic diagram depicting various molecular players modulated by miRNAs in PD, HD, and ALS [83,84,96,99,118,139,140]. The miRNA target gene schematic diagram highlights some of the important genes whose levels are post transcriptionally modified miRNAs, leading to alteration of various vital cellular functions of the neurons. This eventually leads to their degeneration.

**Table 1 genes-13-00425-t001:** The table summarizes some of the important bioinformatics tools widely used for predicting miRNA targets in various organisms. It highlights the important key features, application-based advantages, and disadvantages of these tools.

Tools	Type + URL	Charachteristics	Advantages	Disadvantages	References
TargetScan	Web-based; http://www.targetscan.org/ (accessed on 25 December 2021)	Parameters include a high probability score means a high Pct score, high aggregate Pct score, higher number of conserved sites, and so on. A high Pct score means there is a high level of conservation which indicate that there is a cut down of poor possibilities from the search engine list, which leaves the highest probability choices.	Most robust tool, because it enables a more complete search at isoform level, it penalizes the less conserved interactions, and its databases are the most up-to-date. It is stricter about the interaction site and considers only the seed region and the 3′ UTR in the search without supporting mismatches. It heavily prioritizes the conservation level of miRNA:mRNA interactions and rejects the interactions in ORFs and 5′ UTR regions as ineffective at inducing repression.	Low sensitivity, not well suited when trying to obtain new interaction sites or sites that do not have a strong selective pressure.	[31,32,33]
miRanda	Web-based and downloadable programs; http://www.microrna.org/microrna/getGeneForm.do (accessed on 25 December 2021).	Parameters like mirSVR score, PhastCons score, seed type, and the miTG score are used in analysis using this tool. The miTG score can range from 0 to 1 and defines the interaction between microRNA and mRNA, the higher the number means that the confidence level is higher. The PhastCons score defines the conserved sequence amongst different species. The mirSVR scores portray the probability of down-regulation which can also act as a cut-off score.	High sensitivity, helps in analyzing non-conserved sites and miRNA recognition elements, which can also comprise CDS interaction sites.	High false positive rates.	[31,34]
RNA22	Web-based; https://cm.jefferson.edu/rna22/ (accessed on 25 December 2021)	Uses pre-existing sequences in recognized mature miRNA.This distinct characteristic can still be functional because the reverse complement of the existing sequence will allow the researcher to discover possible microRNA target sites in the sequence provided in the search. When this possible microRNA target site has been identified, the target microRNA will be discoverable.	Ability to identify microRNA target sites that may not be a part of a conserved sequence in organisms that are close in terms of phylogenetics. An RNA22 does not filter out potential microRNA target sites based on cross-organism conservation boundaries. RNA22 tool is an option for searching for new miRNA:mRNA interactions, because its predictions are independent of the state of conservation and also includes interactions along the entire mRNA (3’ UTR, 5’ UTR, and CDS regions).	It can generate a lot of false positives.	[35]
miRBase	Web based; https://www.mirbase.org (accessed on 25 December 2021)	Helps in the identification of boundary predictions through the view of their locations on miRNA primary transcripts that have been entered into the database.	MiRBase allows us to identify predictions of boundaries through the view of their locations on miRNA primary transcripts that have been entered into the database.		[36]
PicTar	Web-based; PicTar http://pictar.mdc-berlin.de/ (accessed on 25 December 2021)	An entry into this algorithm will provide comprehensive information regarding microRNA target predictions in the *Drosophila* species, vertebrates, as well as human microRNA targets that have been apparent as co-expressed but not as a conserved sequence. An example of this co-expression could be the mRNA and microRNA being expressed in the same tissue.	The excellent success rate in predicting targets for single microRNAs and for combinations of microRNAs provides comprehensive information regarding microRNA target predictions in *Drosophila*, vertebrates, and human microRNA targets that have been apparently coexpressed and do not bear conserved sequence.		[37]
PITA	Web-based; http://genie.weizmann.ac.il/pubs/mir07/mir07_prediction.html (accessed on 25 December 2021)	Uses target-site accessibility for miRNA target prediction. PITA identifies a potential site by seed match criteria, then takes into account site accessibility by computing a free energy score. Target-site abundance is calculated by combining site accessibility scores for the same miRNA to identify a total interaction score for the miRNA and UTR.	PITA can predict which miRNA might target a user-provided UTR sequence. This feature is advantageous for the user who wishes to evaluate the 3′ UTR of a novel gene or the 5′ UTR of a gene of interest.	Predictions are based on miRNA sequences from miRBase version 11, a very old version of miRBase without recent updates.	[37,38]

**Table 2 genes-13-00425-t002:** The table summarizes some of the other important miRNAs with altered levels in AD cases. Some of the miRNAs have experimentally validated targets, known to be important in regulating various pathological processes in the neurons leading to AD. On the other hand, certain miRNAs have been shown to have altered levels in blood plasma, cerebrospinal fluid, or postmortem brain tissues in AD patients. However, their molecular targets have not been experimentally validated to yield the cellular function disrupted to cause pathological hallmarks of the disease.

miRNA	Level Changes	Molecular Targets	Pathologic Process	References
miR-15b	Downregulated	BACE1 and APP	Abeta accumulation, Tau toxicity and cell death.	[59]
miR-93	Downregulated			[65]
miR-127-3p	Downregulated		Cell death.	[46]
miR-214	Downregulated	Atg12	Autophagy.	[60]
miR-let-7f-5p	Downregulated	Caspase 3	Cell death or apoptosis.	[61]
miR-124	Downregulated	BACE1	Synaptic dysfunction.	[62]
miR-188	Downregulated	BACE1	Abeta accumulation.	[63,64]
miR-219	Downregulated	Tau or MAPT (Microtubule associated protein Tau)	Tau toxicity.	[42]
miR-342-3p	Upregulated	Activation of JNK-MAPK cascade	Abeta accumulation.	[46,67]
miR-455-3p	Upregulated	APP, NGF, USP25, PDRG1, SMAD4, UBQLN1, SMAD2, TP73, VAMP2, HSPBAP1, and NRXN1	Abeta accumulation.	[47]
miR-146a	Upregulated	NF-kB	Inflammation.	[48,49]
miR-34a	Upregulated	ADAM10, NMDAR 2B and SIRT1	Cell death, Tau phosphorylation-dephosphorylation, APP metabolism.	[50,68]
miR-30a-5p	Upregulated	BDNF	Synaptic dysfunction.	[51]
miR-206	Upregulated	BDNF	Synaptic dysfunction.	[52]
miR-128	Upregulated	PPAR-γ	Tau toxicity.	[53]
miR-106b	Upregulated	Fyn	Apoptosis, Tau phosphorylation.	[54,66]
miR-330	Upregulated	VAV1 via the MAPK pathway	Abeta production, mitochondrial dysfunction.	[55]
miR-195	Upregulated	BACE1	Abeta accumulation.	[56]
miR-200	Upregulated	S6 kinase B1, mTOR	Modulate Abeta secretion and Abeta dependent cognitive impairment by altering insulin signalling.	[57]
miR-9	Upregulated	APP, UBE4B	Abeta accumulation and inflammation.	[58,69]

**Table 3 genes-13-00425-t003:** In this table the various methods of restoration of miRNAs have been listed with their advantages and disadvantages. The restoration methods include both means of increasing the expression of miRNAs as well as their knockdowns. Based on the therapeutic requirement, these strategic methods have been designed to effectively modulate the levels of disease-related miRNAs to ameliorate the disease phenotypes or symptoms.

miRNA Restoration, [References]	Characteristics	Strengths	Weaknesses
Anti-miRs oligonucleotides (AMOs), [176,179,180]	Synthetic, single-stranded antisense RNA oligonucleotides designed, to be complementary to the target miRNA.They bind to miRNAs inside the RISC complex.	(1) Suppress the function of a specific miRNA; (2) Broadly effective, (3) Used in vitro and in vivo to discover gene function, and some AMOs are being tested in clinical trials.	(1) Poorly suited to in vivo applications due to poor cell membrane penetration and degradation by nucleases, (2) Chemical modifications are required to increase resistance to serum nucleases, to enhance their binding affinity, biostability, specificity for the target miRNAs, and to improve their entry into the cell, (3) Limited tissue distribution when administered in the absence of a carrier, are taken up by the liver and kidney and rapidly excreted in the urine.
Antagomirs, [174,175]	Silence endogenous miRNAs, Chemical modification helps to increase their binding to a target miRNA and/or resistance to degradation by nucleases.	(1) Fully complementary to mature miRNAs—competitive inhibition, (2) Can be efficiently delivered into the cytoplasm of a cell.	(1) High dose required for in vivo inhibition (~80 mg/kg) to achieve the same efficacy as other AMO strategies, increases the risk of off-target effects, (2) Cannot cross the blood-brain barrier, require direct injection into the brain.
miRNA sponges, [177,179,180]	Synthetic RNA molecules have multiple tandem repeats of specific miRNA-binding sites.	(1) Can stably interact with the corresponding miRNA and prevent its interaction with its target mRNAs, (2) Can interfere with the activity of all closely related miRNAs within a family that share the same ‘seed sequence’, (3) Can be stably integrated into chromosomes, designed to be drug inducible or controlled by promoters whose expression is restricted to a specific cell type, tissue, or developmental stage, (4) Have demonstrated more effective inhibition of miRNA function compared to other methods such as antagomirs.	(1) Restricted utility in vivo-their usage has been limited to transgenic animals in which the sponge mRNA is overexpressed in target tissues, (2) Efficiency depends on miRNA affinity and on sponge:miRNA stoichiometry, (3) Sponges appear to be degraded by Argonaute 2 in the RISC and thereby hold weaker inhibitory activity.
Locked nucleic acid (LNA) anti-miRs, [174,177,178,179,180]	Short, single-stranded LNA-modified oligonucleotides. These anti-miR reagents have an extra methylene bridge connecting the 2′-O atom and the 4′-C atom ‘locks’ the ribose ring in a C3′-endo or C2′-endo conformation.	(1) Small size, potency, stability, and specificity provided by the LNA modifications, enable delivery possible without vehicle systems. (2) Exhibit higher thermal stability and superior hybridization with their RNA target molecules, (3) Display higher aqueous solubility and increased metabolic stability for in vivo delivery, (4) They can inhibit all members of the same miRNA family or of several miRNA families that share the same ‘seed region,’ inducing a consequent upregulation of their direct targets.	Only moderate efficiency for miRNA inhibition, possible because of the tendency of LNA oligonucleotides to form dimers with exceptional thermal stability.
miR-Mask, [174,177]	Single-stranded 2′-O-methyl-modified antisense oligonucleotides with locked 5′ and 3′ ends that are complementary to the miRNA-binding sites in the 3′UTR of target mRNA.	They mask the target mRNA from the endogenous miRNA and thus prevent its suppression. This specific mechanism reduces the off-target effects and is highly target-specific.	
miRNA expression vectors, [174,180]	Plasmid or viral expression vectors with strong promoters.	Restoration of the expression and function of a specific miRNA, Viral delivery of miRNAs can be optimized to achieve specific and continuous expression level, Evidence of high transduction efficiency with low toxicity.	Less efficient due to transcription of DNA to miRNA precursors, and the need of their delivery to the nucleus, Side effects reported due to overexpression of shRNA in rats leading to hepatotoxicity, organ failure, and death, Argonaute and exportin-5 limit the amount of exogenous miRNA that a cell can tolerate.
miRNA mimics, [174,180]	Small, chemically modified double-stranded miRNA molecules, that undergo intracellular processing by RISC machinery into single-strand forms.	Small, chemically modified double-stranded miRNA molecules, that undergo intracellular processing by RISC machinery into single-strand forms, Increases the levels of a miRNA that is lost during disease progression.	Systemic delivery can result in uptake by non-target tissues resulting in potential off-target effects. They can induce nonspecific interferon response through Toll-like receptors.

**Table 4 genes-13-00425-t004:** These tables list the various methods for drug delivery in the central nervous system, using non-viral vectors. The tables also highlight the strengths and weaknesses of each of these methods of delivery, aiding in understanding the best strategy for miRNA introduction to the brain for effective therapy.

Non-Viral Delivery Systems; [References]	Strengths	Weaknesses
Liposomes; [109,112,114,120,121,122,123,124,125,126]	(1) Reduce the efflux of drugs out of the BBB. (2) Entrap both hydrophilic and lipophilic drugs. (3) Weakly immunogenic and biodegradable. (4) Protects the encapsulated therapeutic agent against rapid enzymatic degradation. (5) High versatility and flexibility in the surface modification with target recognition molecules. (6) Minimizing unwanted inactivating effects of the body and improving the bio-distribution of the encapsulated drug to specific cells. (7) Low elimination by the liver and spleen, increases the circulation time of therapeutic agents in the bloodstream and improves the bioavailability of encapsulated molecules for therapeutic action.	(1) Traditional liposomes have low transfection efficiency into cells due to their lack of surface charges. (2) Nonspecific uptake, and unwanted immune response. (3) Usually heterogeneous in size owing to interactions between water molecules and the hydrophobic groups of lipids, and sometimes the large size of the liposomes produces micro-embolisms giving a false impression of brain uptake. (4) Conventional liposomes, composed of cholesterol and phospholipids, suffer from high plasma clearance and low transport across BBB.
Polymeric Nanoparticles; [109,112,114,120,121,122,123,124,125,126,127]	(1) High biodegradability, biocompatibility, non-allergic, low immunogenicity, and lack of or low cytotoxicity, higher stability in biological fluids and protection of the RNA against degradation by RNases, reduced nonspecific biodistribution, encapsulate large amounts of genetic material (high drug-binding capacity), and high delivery efficacy, facilitate the cellular uptake via endocytosis.	(1) High cellular toxicity.
Lipoplexes: Formed by cationic liposomes that self-assemble in the presence of RNA due to the electrostatic interaction between the positively charged lipids and the negatively charged RNA molecules. [109,114,118,119,121,123,124,125,126]	(1) Efficient internalization of RNA via membrane fusion with the host cell, and high rate of endosomal release of RNA after entering the cell.	(1) It can induce inflammatory effects and unwanted interaction with negatively charged serum proteins, which can lead to opsonization and clearance of the lipoplex.
**Nonviral Delivery Systems; [References]**	**Strengths**	**Weaknesses**
Exosomes; [112]	(1) Derived from intraluminal vesicles and are released from the plasma membrane; contain proteins, lipids, and miRNAs that can mediate various signaling functions; CNS-derived exosomes are released into physiological biofluids such as CSF and blood. (2) Exosomes can be used as diagnostic tools and have reduced immunogenicity and toxicity.	(1) Possible effects of nucleic acids and proteins derived from dendritic cells and carried with the exosomes on the target cell need to be further explored.
Dendrimers (Composed by repetitive units of branched molecules; ability to control their structure); [112,114,123,124,126]	(1) High versatility to incorporate multiple molecules in the peripheral end groups. (2) Improve solubility, pharmacokinetics, and biodistribution of the therapeutic agents. (3) High loading capacity and transfection efficiency. (4) Low toxicity and immunogenicity; triggering endosomal escape and release RNA into the cytoplasm. They are cleared rapidly by the bloodstream, preventing ‘long-term’ accumulation in nontargeted organs, such as kidneys, lungs, and liver, reducing potential side effects.	(1) Controlled drug release and high drug loading still remain challenges with dendrimers. (2) Their cytotoxicity increases proportionally with the generation number.
Cyclodextrins; [112,114,119,123,124,125]	(1) Naturally derived materials with the ability to deliver therapeutic agents across the BBB. (2) Cyclodextrins have been investigated intensely in the targeted delivery of small therapeutic molecules due to their nontoxicity and not producing immune stimulation.	
Polymeric micelles: Amphiphilic copolymers composed by a hydrophobic core and hydrophilic surface [112,119,122,123].	(1) Easy to formulate, incorporated at different sites in micelles. (2) Small particle size that allows escaping from the reticuloendothelial system. (3) Enhanced drug solubility, drug pharmacokinetics, and bio-distribution; high physical stability.	(1) Enhanced penetration for a number of useful drugs, using this non-viral delivery system would also open the BBB to potentially toxic substances.

**Table 5 genes-13-00425-t005:** This table highlights some of the widely used viral vectors for gene delivery into the CNS and the advantages and disadvantages in their method of delivery [197,198,199].

Viral Delivery Systems	Strengths	Weaknesses	References
Adeno-associated virus (AVV)	Viral vectors are currently being used more frequently in the CNS. They are neurotrophic, can exist stably with a low rate of genomic integration, exhibit no pathogenicity or cytotoxicity, can be manufactured at high titers and at high purity, high efficiency in vivo delivery.	Small packaging capacity, leads to severe limitations on the therapeutic cargo size.	[197,198,199]
Adenoviral	The transgene does not integrate into the host genome but remains episomal, leading to stable and sustained expression in the brain for at least up to a year; direct infusion into brain parenchyma results in gene transfer to a broad range of cell populations, including neurons, astrocytes, microglia, and oligodendrocytes.	Small packaging capacity, which places severe limitations on the therapeutic cargo size.	[197,198,199]

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
