# Peer review of "Role of miRNAs in Neurodegeneration: From Disease Cause to Tools of Biomarker Discovery and Therapeutics"

_genes, 2022, doi:10.3390/genes13030425_

Round 1
Reviewer 1 Report
This review by B. Roy and colleagues attempts to summarize in a fairly compact format what has evolved over the last decade to become a rather wide field. The authors manage to accomplish their objective reasonably well; the review is structured and overall quite readable.
I do, however, find that the comprehensiveness of sources varies widely between different parts of the text. Thus, while the authors make use of as many as nine or ten references to make a point (lines 593, 760), sources are sparse for the wealth of knowledge regarding miR-132 (starting on line 332). The involvement of this microRNA in neural biology and pathology has been extensively studied since 2009, and the authors should expand the discussion beyond the group of Hebert et al (who are quoted 6 times) to include other groups responsible for the major findings, not only for miR-132 but other microRNAs as well, and whose publications are not quoted at all.
Besides this necessary re-balancing of sources used for the review, I'm not certain that I see the importance of presenting miRNA target prediction algorithms as a figure (Fig. 1), in the overall context of the review.
Additionally, Table 2 needs work. It is not currently readable due to small font size, and instead of blocks of text it should be summarized into brief points which should also be supported by references.
Finally, I would also recommend a minor round of proofreading, for language and syntax (e.g. lines 89-90: "The transcriptional regulation of miRNA sometimes adopt feedback loop").
Author Response
This review by B. Roy and colleagues attempts to summarize in a fairly compact format what has evolved over the last decade to become a rather wide field. The authors manage to accomplish their objective reasonably well; the review is structured and overall quite readable.
1.I do, however, find that the comprehensiveness of sources varies widely between different parts of the text. Thus, while the authors make use of as many as nine or ten references to make a point (lines 593, 760), sources are sparse for the wealth of knowledge regarding miR-132 (starting on line 332). The involvement of this microRNA in neural biology and pathology has been extensively studied since 2009, and the authors should expand the discussion beyond the group of Hebert et al (who are quoted 6 times) to include other groups responsible for the major findings, not only for miR-132 but other microRNAs as well, and whose publications are not quoted at all.
We have added a table (Table 2) of the most relevant microRNAs involved in AD, with their targets, extent of changes in AD, and the cellular mechanism underlying the pathogenicity (references included). Due to the already extensive length of the manuscript, describing every microRNA involved in AD in details in the text will be difficult. However, we hope that the table will help our readers get a relevant understanding how these specific microRNAs modulate the disease.
2.Besides this necessary re-balancing of sources used for the review, I'm not certain that I see the importance of presenting miRNA target prediction algorithms as a figure (Fig. 1), in the overall context of the review.
We have removed the Figure 1 and added a table of the various bioinformatics tools stating their advantages and disadvantages, and the scenarios in which each one can be used.
3.Additionally, Table 2 needs work. It is not currently readable due to small font size, and instead of blocks of text it should be summarized into brief points which should also be supported by references.
We have modified the table.
- Finally, I would also recommend a minor round of proofreading, for language and syntax (e.g. lines 89-90: "The transcriptional regulation of miRNA sometimes adopt feedback loop"). We have taken care of language and syntax.

Reviewer 2 Report
Manuscript "Role of microRNAs in neurodegeneration: from disease cause to tools of biomarker discovery and therapeutics "is an interesting study of issues related to the importance of microRNA in the diagnosis and therapy of neurodegenerative diseases. The authors thought about the scope of topics quite well, but the manuscript is organizationally rather poorly prepared, which makes its reception difficult. Overall, the manuscript is good, although the authors did not avoid the following errors:
- There is no introduction (in the form of a paragraph) emphasizing the need to search for new miRNAs as biomarkers and molecules with therapeutic potential. Are the current diagnostics and therapy ineffective? too late unbelievable? too little sensitive? ... etc.
- It seems to me that the subsection: "MicroRNA biogenesis and dysregulation" is too extensive and not necessarily needed in this study. Excellent reviews of biogenesis only have been published so far, hence such a detailed discussion is, in my opinion, unnecessary and adds little to the key part of the work. This section also has a rather strange structure: with lots of bullets (including double and triple ":") and enumerations. Without a figure, it is quite illegible and difficult to read. In my opinion, it should be shortened or completely removed or redrafted into liquid text.
- Line 54 - the first name is listed, not the surname of the person who described line-4 for the first time.
- “MicroRNAs (miRNA) are small, non-coding RNA molecules transcribed from RNA polymerase II and III ”- this sentence appears too late (line 189): because it contains both the definition, introduces the name and repeats the information from the section above ....
- There are more of these types of errors / inaccuracies throughout the manuscript. Please read and edit the text carefully.
- The subsection "Tools for detecting microRNA targets" in the first part describes the mechanisms of action of miRNAs, and then the tools. This subsection has disproportionately fewer citations compared to the chapters above - large fragments of the text have only single citations (analogically lines 712-740). There are no links with the names of the programs (e.g. links to websites); Moreover, it is too technical and adds little to the very topic of the role of miRNAs in neurodegenerative diseases. In my opinion, I should be shortened and redrafted.
- Line 194 - there are many more miRNA inhibition strategies, which should be better described as these strategies are used in the design of new drugs.
- Figure 1 is illegible and not very informative. Contains only the tool names listed. It would be better to replace it with a table containing a summary of the characteristics of the capabilities of individual programs, the scope of their application, advantages or disadvantages….
- Line 334 "Herbert et al.", firstly, the correct spelling of this researcher's name is "Hébert", and secondly, it is a strange practice to mention the name of the third and not the first author in the text, so it should be: "Hernandez-Rapp et al. . " Similarly, in line 435 there is a citation of "Packet et al. and colleagues ”, while the authors of the work are Ballas et al. !!! Packet's name is not even on the co-authors list !!! Line 447 "Buckley & Johnson wrote", and it should be: "Johnson et al." Because the work has more than two authors. In turn, in line 499 we have "Doxakis and coworkers", but the author is only 1 !!! I am not listing any more, although there are many errors of this type - please correct the bibliography in the text carefully! By the way, there is no uniformity in the style of citation, as the expressions 'et al.', 'and co-workers' and 'at collegues' appear both.
- Where do the data on the basis of which figures 2 and 3 have been prepared come from? There are no citations of the articles from which the data comes. Similarly, on the basis of which articles were Tables 1 and 2 prepared? Also, the tables have a strange format and are poorly prepared. They should contain summary and headline information, not full sentences randomly combined, as if taken from various articles.
- In the subsection "MicroRNA in biomarker study", part of the information on the description of miRNAs in individual diseases should be moved to the subsections above, where these diseases are described. Reception of information would be easier once sorted out.
- Citations in the text are poorly prepared (numbering in multiple brackets) - from line 63 to the end of the text.
- The text contains numerous punctuation, stylistic and lexical errors (e.g. "MicroRNA Biogenesis dysregulation", "invivo and invitro models", "Huntington gene (Htt) mutant", "8mer, 7 mer", "Targets in Parkinson's Disease ( PD): Parkinson's Disease (PD) "," verview of the Available Therapeutic Strategies for optimal "etc.).
- Abbreviations and acronyms are used earlier than they are explained and developed, and some have not been explained at all, so the reader has to search for their meaning. Eg. gene names - some are listed as acronyms, some have explanations later on these of the text and some of them have the full name and the abbreviation given immediately…. of the text and some of them have the full name and the abbreviation given immediately…. - punches are common and too varied, e.g. line 83-85 and line 370-371 I am asking the Authors for an appropriate explanatory comment or to make corrections in the text of the study.
- Bullet points are widespread and too varied, e.g. line 83-85 and line 370-371.
I am asking the Authors for an appropriate explanatory comment or to make corrections in the text of the manuscript.
Author Response
- There is no introduction (in the form of a paragraph) emphasizing the need to search for new miRNAs as biomarkers and molecules with therapeutic potential. Are the current diagnostics and therapy ineffective? too late unbelievable? too little sensitive? ... etc.: We have added this in the introduction. Please also read the therapeutics section. We have added some more latest research in terms of clinical trial hurdles of microRNA etc.
- It seems to me that the subsection: "MicroRNA biogenesis and dysregulation" is too extensive and not necessarily needed in this study. Excellent reviews of biogenesis only have been published so far, hence such a detailed discussion is, in my opinion, unnecessary and adds little to the key part of the work. This section also has a rather strange structure: with lots of bullets (including double and triple ":") and enumerations. Without a figure, it is quite illegible and difficult to read. In my opinion, it should be shortened or completely removed or redrafted into liquid text. We have shortened the length of this subsection and presented only the relevant points. We have also removed bullet points and completely restructured this section.
- Line 54 - the first name is listed, not the surname of the person who described line-4 for the first time.
We have added the last name of the corresponding author in the Introduction section.
- “MicroRNAs (miRNA) are small, non-coding RNA molecules transcribed from RNA polymerase II and III ”- this sentence appears too late (line 189): because it contains both the definition, introduces the name and repeats the information from the section above .... We have removed this from the tools for target prediction and added to the introduction subsection.
- There are more of these types of errors / inaccuracies throughout the manuscript. Please read and edit the text carefully. We have taken care of the typos.
- The subsection "Tools for detecting microRNA targets" in the first part describes the mechanisms of action of miRNAs, and then the tools. This subsection has disproportionately fewer citations compared to the chapters above - large fragments of the text have only single citations (analogically lines 712-740). There are no links with the names of the programs (e.g. links to websites); Moreover, it is too technical and adds little to the very topic of the role of miRNAs in neurodegenerative diseases. In my opinion, I should be shortened and redrafted.
We have added links or websites where these programs can be accessed. We have included this section of tools for target prediction as these tools help in hinting towards some main targets of individual microRNAs, and set a platform for further experimental validation.
- Line 194 - there are many more miRNA inhibition strategies, which should be better described as these strategies are used in the design of new drugs. We have added
- Figure 1 is illegible and not very informative. Contains only the tool names listed. It would be better to replace it with a table containing a summary of the characteristics of the capabilities of individual programs, the scope of their application, advantages or disadvantages…. We have removed Figure 1 from the review. We have added advantages and disadvantages of each of these target prediction tools in the text.
- Line 334 "Hébert et al.", firstly, the correct spelling of this researcher's name is "Hébert", and secondly, it is a strange practice to mention the name of the third and not the first author in the text, so it should be: "Hernandez-Rapp et al. . " Corrections have been made.
Similarly, in line 435 there is a citation of "Packet et al. and colleagues ”, while the authors of the work are Ballas et al. !!! Packet's name is not even on the co-authors list !!! Corrections have been made.
Line 447 "Buckley & Johnson wrote", and it should be: "Johnson et al." Because the work has more than two authors. In turn, in line 499 we have "Doxakis and coworkers", but the author is only 1 !!! I am not listing any more, although there are many errors of this type - please correct the bibliography in the text carefully! By the way, there is no uniformity in the style of citation, as the expressions 'et al.', 'and co-workers' and 'at collegues' appear both.Corrections have been made.
- Where do the data on the basis of which figures 2 and 3 have been prepared come from? There are no citations of the articles from which the data comes. Similarly, on the basis of which articles were Tables 1 and 2 prepared? Also, the tables have a strange format and are poorly prepared. They should contain summary and headline information, not full sentences randomly combined, as if taken from various articles. We have added citations to both the figures. This figure is generated from information available from these articles.
- In the subsection "MicroRNA in biomarker study", part of the information on the description of miRNAs in individual diseases should be moved to the subsections above, where these diseases are described. Reception of information would be easier once sorted out. We have kept the two sections separate since the articles cited in biomarker section spans research conducted meant or relevant for microRNA biomarker studies only. The other sections had citations focussing on target identification to understand the disease mechanisms.
- Citations in the text are poorly prepared (numbering in multiple brackets) - from line 63 to the end of the text.We have made changes wherever applicable.
- The text contains numerous punctuation, stylistic and lexical errors (e.g. "MicroRNA Biogenesis dysregulation", "invivo and invitro models", "Huntington gene (Htt) mutant", "8mer, 7 mer", "Targets in Parkinson's Disease ( PD): Parkinson's Disease (PD) "," verview of the Available Therapeutic Strategies for optimal "etc.). Changes have been made.
- Abbreviations and acronyms are used earlier than they are explained and developed, and some have not been explained at all, so the reader has to search for their meaning. Eg. gene names - some are listed as acronyms, some have explanations later on theseof the text and some of them have the full name and the abbreviation given immediately…. of the text and some of them have the full name and the abbreviation given immediately…. - punches are common and too varied, e.g. line 83-85 and line 370-371 I am asking the Authors for an appropriate explanatory comment or to make corrections in the text of the study. Some genes do not have full forms. We have tried to quote them in the manner similar to the references cited.
- Bullet points are widespread and too varied, e.g. line 83-85 and line 370-371. We have removed the bullet points.
I am asking the Authors for an appropriate explanatory comment or to make corrections in the text of the manuscript.

Round 2
Reviewer 1 Report
The manuscript appears to be improved. However, I would appreciate seeing a clean version without markup, as the current file is difficult to read.
Author Response
please find it in the attachment

Reviewer 2 Report
The Authors of the manuscript entitled "Role of microRNAs in neurodegeneration: from disease cause to tools of biomarker discovery and therapeutics" quite selectively approached the revision of the reviewed manuscript. There was no in-depth and detailed examination of the text of the manuscript in accordance with the outlined guidelines. I have included examples of errors in the detailed guidelines, and I hoped that the Authors would correct the entire text in accordance with them. I have the impression that the Authors applied some of the suggested changes, but did not comment on why they did not implement the others (except for the names of the genes). Unfortunately, from the very first sentences, the manuscript is marked by numerous editing errors, which give the impression that the article is prepared sloppy. I would have to repeat a number of remarks from the previous review, including incorrect abbreviations, incorrect punctuation, incorrect use of paragraphs, capital letters in the middle of a sentence, errors in the names of the authors cited, missing citation ... etc. I will not list here again examples of such errors (you can return to the previous review) - I recommend a careful analysis of the text. I would like to discuss two aspects in detail: 1. The strength of a review analysis should be an excellent knowledge of the literature on the subject, as well as a skilful presentation of literature data. While the Authors managed to substantively deal with the presentation of the content, the use of citations raises big doubts, because: - there are errors in the names of the authors, the co-author is confused with the first author, the list of references includes items that have not been quoted in the text, the order of citations is random (e.g. the introduction starts at item 8, and item 1 is several pages further), the introduction ends with the citations of item 25, and in the next subsection they start with item 44), while in Table 1 the references are incorrect (the sources are incorrectly cited - the contents of the tables do not match the text of the cited articles), different citation and formatting styles are used (both numeric and textual), the reference list is inconsistent with the citations used in the manuscript ... 2. Tables are illegible and incorrect: they should contain summary statements, comparisons and summaries - easy to trace; there are errors in their numbering and references to them (e.g. on page 17 there is table 4, and in the text there is a reference to table 2). Table 1 is too extensive - the authors use full-sentence descriptions unnecessarily. In addition, there are incorrect references to the literature in this table. In Table 2, the target mRNA for individual markers should be added (they are to be found in the databases mentioned by the authors in Table 1), the list of references should be the last column (besides, the style of giving the refence needs to be harmonized with Table 1), and some of the references cited are missing in the list at the end of the manuscript, which makes it difficult to verify their correctness. The column "Pathological process" should be completed in full, since the authors, according to the legend of this table, present miRNAs involved in the pathogenesis of AD in it. If the authors are not able to assign a specific role to a given miRNA molecule, it should not be included in the table below. Tables 3 and 4, similar to Table 1, are too extensive and poorly compiled. Moreover, they do not contain references to the literature. The legend of table 5 contains information about continuation (unnecessary), because it is either table 5 or continuation of table 4. Besides, there are no references again. The text, on the other hand, lacked links to tables. Summarizing, the Authors must carefully trace and correct the text in order to facilitate the reception of the content by potential readers. The manuscript should become more readable and the citations should be clear.Author Response
Dear Editor,
- The concerns of reviewer 2 that some of the authors were co-authors and we did not cite the second co-authors, has been corrected.
- Paragraphs have been separated.
- We have 2 figures and 5 tables, with table 4 having 2 parts, a and b. This was done to make the contents legible. I have seen some reviews where authors have continued tables to 2 pages. All figures and tables have a heading and few lines indicating the content. The tables have a font size that is legible and we used Arial 10 (which is greater than genes table’s permissible font size of 8) for the text size.
- All tables have references added in the tables itself. References have been added also in the legend of the two figures.
- Additionally, we have added the full forms of all the genes in brackets.
- All syntax and grammatical errors have been corrected using the software, Grammarly Check.
- We have revised the manuscript as per as the instructions.
- Additional miRNAs modulating Tau could not be added in the text owing to the already extensive length of the manuscript. However, we have added Table 2 listing additional these miRNAs, with their individual targets, mode of pathogenesis and cited the necessary references.
Round 3
Reviewer 1 Report
The authors have addressed prior comments reasonably well, and I recommend acceptance.
Reviewer 2 Report
I would like to thank the authors for their contribution to improving the quality of the manuscript. I still believe the tables are incorrectly prepared. In my opinion, they should have a compact layout, and the content should be "slogan / concise" and "summary". In their current form, they are overloaded with content and therefore illegible. Please also put the references in the last column of the table (so that all tables have a uniform format).